# A Scalable Implementation of Anonymous Voting over Ethereum Blockchain

**DOI:** 10.3390/s21123958

**Published:** 2021-06-08

**Authors:** Jae-Geun Song, Sung-Jun Moon, Ju-Wook Jang

**Affiliations:** Department of Electronic Engineering, Sogang University, Seoul 04107, Korea; skj1080@sogang.ac.kr (J.-G.S.); ckop1@nate.com (S.-J.M.)

**Keywords:** electronic voting, e-voting, blockchain, Ethereum, scalability

## Abstract

We considered scalable anonymous voting on the Ethereum blockchain. We identified three major bottlenecks in implementation: (1) division overflow in encryption of voting values for anonymity; (2) large time complexity in tallying, which limited scalability in the number of candidates and voters; and (3) tallying failure due to “no votes” from registered voters. Previous schemes failed at tallying if one (or more) registered voters did not send encrypted voting values. Algorithmic solutions and implementation details are provided. An experiment using Truffle and Remix running on a desktop PC was performed for evaluation. Our scheme shows great reduction in gas, which measures the computational burden of smart contracts to be executed on Ethereum. For instance, our scheme consumed 1/53 of the gas compared to a state-of-the-art solution for 60 voters. Time complexity analysis shows that our scheme is asymptotically superior to known solutions. In addition, we propose a solution to the tallying failure due to the “no vote” from registered voters.

## 1. Introduction

### 1.1. Motivation

Electronic voting (e-voting) should satisfy somewhat conflicting requirements: transparency and anonymity. Transparency demands consistent and temper-resistant tallying while anonymity prohibits exposing who votes for which candidate [1]. 

The Ethereum [2] blockchain platform is a good candidate for electronic voting. First, no trusted third party is necessary. It would be easy to construct an electronic voting system with a server (a typical trusted third party). However, the server may become a single point-of-failure. Blockchain replaces the server by a group of independent nodes, eliminating the single point-of-failure.

Second, the hashing and chaining of blocks in blockchain make tempering or modifying voting data mathematically impossible [3,4].

Third, Ethereum blockchain allows us to program smart contracts, which are executed automatically and forcefully by all participating nodes, guaranteeing precision [5].

However, we identified three major bottlenecks when attempting to implement previous systems or protocols [1,6,7] on Ethereum: (1) division overflow in computing encrypted voting values for anonymity (cycle group theory involved); (2) large time complexity in tallying, which severely limited scalability in the number of candidates and voters; and (3) tallying failure due to “no votes” from registered voters. Previous schemes failed at tallying if one (or more) registered voters did not send encrypted voting values. 

In this paper, we address the three major bottlenecks by providing algorithmic solutions as well as implementation details. An experiment using Truffle and Remix running on a desktop PC was performed for evaluation. 

Our scheme shows great reduction in gas [8], which measures computational burden of smart contracts to be executed on Ethereum. For instance, our scheme consumed 1/53 of the gas compared to a state-of-the-art solution for 60 voters.

Time complexity analysis shows our scheme is asymptotically superior to known solutions. In addition, we propose a solution to the tallying failure due to “no votes” from registered voters. 

### 1.2. Previous Work

McCorry, Shahandashti, and Hao [1] (OVNet) presented the first implementation of decentralized and self-tallying anonymous voting on blockchain. The OVNet is a boardroom scale voting protocol that is implemented as a smart contract in Ethereum.

The OVNet provides maximum voter privacy, as an individual vote can only be revealed by a full-collusion attack that involves compromising all other voters; all encrypted voting data are publicly available; and the protocol allows the tally to be computed without requiring a tallying authority.

However, we find the scheme limited in two aspects; (1) it only allows a yes/no vote for a single candidate; (2) it is not scalable in the number of voters. Hao, Ryan, and Zielinski [6] devised an internet voting protocol for multiple candidates. However, it is presented as a protocol not meant for blockchain implementation.

Shahzad and Crowcroft [9] proposed a framework based on adjustable blockchain that can apprehend problems in the polling process, the selection of the suitable hash algorithm, the selection of adjustments in the blockchain, the process of voting data management, and the security and authentication of the voting process. However, it does not address scalability issues in implementation on a specific blockchain platform.

K.M. Khan, Arshad, and M.M. Khan [10] investigated performance constraints for a blockchain-based secure e-voting system. They focused on factors such as block generation rate, transaction speed, and block size, which play an important role in determining the performance of a general blockchain. Their research is useful for general blockchain systems. However, it does not address problems that can only occur on the Ethereum network.

Dimitriou [11] also proposed a model for blockchain-based voting, which reduces a large amount of computation and communication overhead. In the model, coercion resistance and receipt-freeness are ensured by means of a randomizer token—a tamper-resistance source of randomness that acts as a black box in constructing the ballot for the user. However, it requires a trusted administrator to register voters and an aggregator to aggregate the voting results.

S. Park, M. Specter, N. Narula, and R.L. Rivest [12] analyze and systematize prior research on the security risks of online and electronic voting, and show that, not only do these risks persist in blockchain-based voting systems, but blockchains may introduce ‘additional’ problems for voting systems. 

Their concern about blockchain-based voting is summarized in the straw man design (‘coins as votes’): the voter registry spends one coin to each public key, corresponding to each voter. To vote, each user spends their coin to the candidate of their choice. After a period, everyone can look at the blockchain, total each candidate’s coins, and select the one with the most coins as the winner.

Using the ‘coins as votes’ design, the authors illustrated problems faced by known blockchain-based voting proposals. We address their concerns by commenting after each problem they suggested.

First, it does not provide a secret ballot. All votes are public, and users can prove (to a third party) how they voted, enabling coercion and vote-selling. We agree with this claim, but this problem is a rather general one for most online-based transactions. A burglar may threaten a person to send money to a foreign account via digital banking that uses public cryptography. We agree that guaranteeing a secure and private human interface to blockchain-based voting remains an important issue. We do not claim we have provided a solution to this issue.

Second, this design relies on users being able to cast their votes on the blockchain in a given time period. The vote tallier cannot wait for all users to spend their coins, because that means a single user could prevent the election from finishing—there must be a cutoff point. An adversary able to influence network connectivity or conduct a denial-of-service attack could prevent users from voting until after the cutoff. Public blockchains, in particular, are limited in throughput and require fees to submit transactions.

During periods of high transaction rates, fees can get quite high, and transactions can be delayed. An attacker willing to spend enough money could flood the blockchain with transactions to drive up fees and prevent users from voting until after the cutoff point has passed.

We address the issue of a single user preventing an election from finishing. We provide a solution for “no votes” from registered voters (lines 433–477). It is one of our contributions in this paper. An attacker willing to spend enough money could flood the blockchain with transactions to drive up fees; solutions depend on situations. If we use a public Ethereum blockchain for small-scale voting, an attacker could spend a large amount of money to disrupt the voting. 

Third, the design only works if the blockchain properly implements the public bulletin board interface. If the blockchain is compromised—e.g., if a majority of the miners or validators collude—then they could sow discord by creating multiple versions of the blockchain to show different people. Alternatively, they could censor certain users’ votes. Several cryptocurrencies have suffered from these types of attacks, where their blockchains were rewritten [13,14,15]. Blockchains are often referred to as “immutable”, but these attacks show that this is not always true in practice, especially for smaller blockchains. We agree with this concern, in a general perspective, but it is beyond the scope of our manuscript.

Fourth, security of this “straw man” hinges on key management. If users lose their private keys, they can no longer vote, and if an attacker obtains a user’s private key, the attacker could then undetectably vote as that user. This concern applies to most online transactions, including digital banking. A more reliable interface, such as a biometric certificate, will be developed in the future in response to this concern. 

Finally, zero-knowledge proof is designed, for a setting, where a party with secret information wants to keep that information secret—which is why they would use zero-knowledge proof—it generally does not prevent a party from voluntarily revealing information. This weakness in privacy, resulting from human interface to ZKP, should be handled in such a manner that nobody can coerce or buy votes from voters. We believe that mail-in votes suffer this weakness. We may install a local booth where privacy is guaranteed. We agree that this will remain a major issue if blockchain-based voting is to be widely employed.

We identified three major bottlenecks in implementing blockchain voting on the most popular platform (Ethereum) and devised schemes to overcome them. 

### 1.3. Contribution of This Paper

(1)We devised and implemented a scalable anonymous voting system for multiple candidates on the Ethereum blockchain. The anonymous voting system by McCorry, Shahandashti, and Hao [1] was also implemented on Ethereum, but only works for a yes/no vote for a single candidate. Hao, Ryan, and Zielinski [6] devised an anonymous voting protocol for multiple candidates, but it is not meant for blockchain implementation.(2)We propose a solution to the division overflow problem in encryption of voting values for anonymity. Voter i publishes $g^{x_{i}y_{i}}g^{v_{i}}$ as an encrypted voting value to guarantee anonymity. The $v_{i}$ represents whom voter i votes for and, thus, should not be revealed. For a yes/no vote in a single candidate case [1], vi is one of {1,0}. For multiple candidates, $v_{i}$ can be encoded to facilitate tallying. To compute $g^{y_{i}}$ voter i need collect $g^{x_{j}}$ for all j(j≠i) and perform a division [1,6]. In regards to huge numbers resulting from encryption, the division may introduce a round-off error. Modular division using the cyclic group [16] can be used to prevent this. Modular division can be performed by a multiplication by the inverse of a multiplicand. However, computing the multiplicative inverse in the modular computation takes $O\left(\log{\left(p\right)}^{2}\right)$ time using the extended Euclidean algorithm. This limits scalability of the scheme since an arbitrarily large p is usually chosen for encryption [17]. To circumvent this, we remove computing the multiplicative inverse. We replace the modular division by a multiplication exploiting a feature of the cyclic group. This removes $O\left(\log{\left(p\right)}^{2}\right)$ time, enhancing scalability of our scheme.(3)We devised a tallying scheme that is superior to known schemes [6,7] in terms of time complexity. The tallying is individually performed in each voter. A tally value is constructed in each voter after it collects all of the encrypted voting values from all other voters. Each voter should resolve this tally value to identify how many votes each candidate has won.The tallying scheme for multiple candidates used in Hao, Ryan, and Zielinski [6] is not scalable if implemented in Ethereum. The tally value grows enormously, even for the modest number of voters or candidates. The tallying requires a division involving the tally number and may result in division overflow. One way to avoid this is to precompute all possible mappings between the tally values and the corresponding tally results (i.e., how many votes each candidate has won). The size of the mapping table is kHn for k candidates and n voters. $O\left(n^{k}\right)$ time and memory space are needed for construction of this table. This can grow very large, even for moderate values of n and k. We devised a new way to encrypt voting values and a new scheme to encode candidates to reduce the time complexity from $O\left(n^{k}\right)$ to $O\left({k\log}_{2}n\right)$.(4)We proposed a solution to the tallying failure due to “no votes” from registered voters. We identified this problem from a previous voting scheme for multiple candidates [6]. The previous tallying scheme was unable to retrieve tally value if one or more registered voters failed to send their encrypted voting values. In reality, we may have this situation more often than not. 

We devised a scheme to allow the remaining voters cooperate, to recover the appropriate tally value excluding the failed voters. First, we show how to recover from the failure of a single voter and then extend our algorithm to multiple failed voters. 

The rest of this paper is organized as follows. Section 2 provides a mathematical foundation. Section 3 identifies three major technical difficulties for implementation on an Ethereum blockchain and presents our solutions. Performance evaluation is presented in Section 4; Section 5 concludes this paper.

## 2. Mathematical Foundation

We provide a mathematical foundation needed in the derivation of our anonymous voting scheme on the Ethereum blockchain. For self-completeness, we include as much detail as possible.

### 2.1. ZKP (Zero-Knowledge Proof)

Zero-knowledge proof (ZKP) is an interactive procedure used for proving (to a verifier) that something that a prover knows is true, without revealing it to the verifier [18]. This is used in anonymous voting systems, to prove that a participant has the private key for a public key published by it. This establishes the ownership of the private key to be used in the encryption of voting values, without revealing the private key.

ZKP is also used to prove that the voter casts a well-formed vote. For example, ZKP ensures that the vote belongs to {0,1} for a single candidate case or {0, 1, 2, …, *k* − 1} for one-out-of-k candidate case. This will prevent potential mischievous voters from disturbing the voting system. Tallying will not be correct if a voter chooses an invalid candidate. This invalid vote should be prevented.

Among the several types of proof, Schnorr Zero-knowledge proof [19] is the most commonly used type. Assume that p is a random prime number and we choose a prime number *q* that satisfies (*p* − 1) mod *q* = 0 as in Equations (1) and (2).
(1)$$p<2^{256}$$
(2)(p−1)modq=0

Let *G*, *g* denote a finite multiplicative cyclic group of *q* and a generator in *G*, respectively, as in Equation (3).
(3)$$g^{q}\mathrm{mod}p=1$$

The (*p*, *q*, *g*) have to be set for performing zero-knowledge proof. The prover performs the following three stages:

Commitment stage: each prover chooses a random value $x_{i}\in$ _R_ℤ *q* as a private key and broadcasts the pairing public key $X_{i}=g^{x_{i}}\;\mathrm{mod}\;p$. Moreover, the prover chooses a private variable *k* ∈ ℤ*_q_* and produces a public variable *K*
$=g^{k}\;\mathrm{mod}\;p$. The prover sends $X_{i}$ and *K* to the verifier without revealing $x_{i}$ or *k*.

Challenge stage: the verifier chooses a random variable *c* (*c*
∈ ℤ*_R_*) to send it to the prover.

Response stage: the prover computes *s* = $\left(x_{i}c+k\right)$mod *q* and sends it to the verifier as in Equation (4).
(4)$$s=(x_{i}c+k)\mathrm{mod}q$$

The verifier performs a verification using (*X_i_*, *K*, *s*) as in (5).
(5)$$Check\;g^{s}=KX_{i}{}^{c}\;(\mathrm{mod}q)$$

### 2.2. Non-Interactive Zero-Knowledge Proof (NIZKP)

At times, the interactive procedure needed in ZKP [19] may be cumbersome or hard to implement. To remove this interactive procedure, non-interactive zero-knowledge proof (NIZKP) was proposed [20].

NIZKP only needs a verifier in the ‘Response Stage’. The prover generates *c* using a hash function on the ‘challenge stage’. Let *H* be a hash function, then *c* is described in Equation (6) as follows.
(6)c=H(K)modq

The ‘response stage’ remains the same as ZKP.

### 2.3. Two-Round Referenda

Assume *G* denotes a cyclic group of prime order *q* where the decision Diffie–Hellman problem is intractable [21]. Let *g* be a generator in *G*, on which *n* participants agree. 

For *i* = 0 to *n* − 1, the following is performed. 

Participant *P_i_* randomly chooses a secret value: $x_{i}\in$ _R_ℤ*q*. We consider the single candidate case first, in which a vote is either ‘yes’ or ‘no’. Then, participants perform the following two-round protocol:

Round 1: *P_i_* publishes $g^{x_{i}}$ and a ZKP for *x_i_*. *P_i_* checks the validity of all the ZKPs and computes Equation (7).
(7)$$g^{y_{i}}=\prod_{j=0}^{i-1}g^{x_{j}}/\prod_{j=i+1}^{n-1}g^{x_{j}}$$

Round 2: *P_i_* publishes $g^{x_{i}y_{i}}g^{v_{i}}$ and a ZKP showing that *v_i_* as shown in Equation (8).
(8)$$v_{i}=\{\begin{array}{cc} 1 & {\mathrm{if}\;P}_{i}\;\mathrm{votes}\;‘\mathrm{yes}’\\ 0 & {\mathrm{if}\;P}_{i}\;\mathrm{votes}\;‘\mathrm{no}’ \end{array}$$
(9)$$\prod_{i=0}^{n-1}g^{x_{i}y_{i}}g^{v_{i}}=g^{\sum_{i=0}^{n-1}v_{i}}$$

Each participant is able to tally the ‘yes’ votes using Equation (9). $\underset{i}{\sum }v_{i}$ is the number of ‘yes’ votes and is denoted by γ. All ZKPs are performed to make sure that ballots are properly cast (i.e., ‘yes’ or ‘no’). Equation (9) holds since $\underset{i}{\prod }g^{x_{i}y_{i}}=1$ from Proposition 1 as in Equation (10).

If γ is a small number, we can easily compute the discrete logarithm of $g^{\gamma }$, by exhaustive search or Shanks’ baby-step giant-step algorithm [22].

**Proposition** **1.***For the x_i_ and y_i_ as defined in the above protocol, we have*$\underset{i}{\sum }x_{i}y_{i}=0$.

**Proof.** By definition $y_{i}=\underset{j<i}{\sum }x_{j}-\underset{j>i}{\sum }x_{j}$, hence
(10)$$\begin{array}{ll} \sum_{i=0}^{n-1}x_{i}y_{i} & =\sum_{i=0}^{n-1}\sum_{j=0}^{i-1}x_{i}x_{j}-\sum_{i=0}^{n-1}\sum_{j=i+1}^{n-1}x_{i}x_{j}\\ & =\sum_{i=0}^{n-1}\sum_{j=0}^{i-1}x_{i}x_{j}-\sum_{j=0}^{n-1}\sum_{i=0}^{j-1}x_{i}x_{j}\\ & =\sum_{i=0}^{n-1}\sum_{j=0}^{i-1}x_{i}x_{j}-\sum_{j=0}^{n-1}\sum_{i=0}^{j-1}x_{j}x_{i}\\ & =0 \end{array}$$

ZKPs are used to ensure participants faithfully follow the protocol, as in [23,24]. In the first round, each participant proves his or her knowledge of the exponent without revealing it. Schnorr’s signature [19] can be used for this purpose. 

In the second round, each participant needs to prove that the encrypted vote is one of {1,0} without revealing which one. For this, they can adapt an efficient technique proposed by Cramer, Damgard, and Schoenmakers (CDS) in [25] (also see [26]). Refer to [25] or [26] for details. □

### 2.4. Extend to Multiple Candidates

The above single-candidate protocol can be extended to accommodate multiple (more than 2) candidates [23,24,25]. Each voter is permitted to select only one candidate. The *k* independent generators $g_{0},g_{1},\ldots ,g_{k-1}$ (one for each candidate) are obtained. The first round is the same as the above. In the second round, participant *P_i_* broadcasts $g^{x_{i}y_{i}}\cdot Q_{i}$ with a ZKP proving that $Q_{i}$ is one of $g_{0},g_{1},\ldots ,g_{k-1}$ (the same CDS technique [25] used). Each participant may perform tallying by computing $\prod_{i}g^{x_{i}y_{i}}\cdot Q_{i}=g_{0}^{c_{0}}\cdot g_{1}^{c_{1}}\cdots g_{k-1}^{c_{k-1}}$ where *c*_0_ to *c**_k_*_−1_ are the number of votes won by *k* candidates 0 to *k* − 1, respectively.

Given *n* voters and *k* candidates, the number of possible tallying results is $\left(\begin{array}{c} n+k-1\\ k-1 \end{array}\right)=O(n^{k-1})$ [27].

This is not scalable if *n* or *k* is large [26]. Another demerit of this scheme is a requirement that the $g_{i}$’s are appropriately independent, as represented in Equation (11), where *N* is the number of voters.
(11)$$\forall c_{i},c_{i}^{\prime }\in N,\prod g_{i}^{c_{i}}=\prod g_{i}^{c_{i}^{\prime }},\forall i\in \{0,\ldots ,k-1\},c_{i}=c_{i}^{\prime }$$

Baudron et al. [7] proposed an improved scheme. The smallest integer, such that $2^{m}>n$ is obtained, and a vote for candidate 0 is encoded as $2^{0}$, for candidate 1 as $2^{m}$, for candidate 2 is $2^{2m}$ and so on. If voter *i* votes for candidate *j* then *v_i_* = *t_j_* as in Equation (12).
(12)$$t_{j}=\{\begin{array}{ccc} 2^{0} & & \mathrm{if}\;\mathrm{Voter}\;i\;\mathrm{votes}\;\mathrm{for}\;\mathrm{candidate}\;0\\ 2^{m} & & \mathrm{if}\;\mathrm{Voter}\;i\;\mathrm{votes}\;\mathrm{for}\;\mathrm{candidate}\;1\\ & \ldots & \ldots \\ 2^{m(k-1)} & & \mathrm{if}\;\mathrm{Voter}\;i\;\mathrm{votes}\;\mathrm{for}\;\mathrm{candidate}\;k-1 \end{array}$$

The tally value is obtained, as in Equation (9). The super-increasing feature is used for this purpose. The scheme requires $2^{m}>n$ for unambiguous resolution. A vote for candidate *j* is coded as $2^{jm}$. Since $n*2^{jm}$ < $2^{\left(j+1\right)m}$, even if all the *n* voters vote for candidate *j*, the value is less than a single vote for candidate *j* + 1. This super-increasing nature guarantees unambiguous resolution, as in Equation (13).
(13)$$g^{\sum_{i=0}^{k-1}v_{i}}=g^{2^{0}\cdot c_{0}+2^{1}\cdot c_{1}+\cdots +2^{(k-1)m}\cdot c_{k-1}}$$

However, this resolution also requires searching over possible *_k_H_n_* combinations, which may grow very large, even for moderate values of *n* or *k*.

### 2.5. Ethereum

Our voting system is implemented on Ethereum, which is the most popular platform for smart contracts. A smart contract [28] is a computer program or a transaction protocol, which is intended to automatically execute relevant actions according to the agreed terms. The goals of smart contracts include the removal of trusted intermediators, arbitrations and enforcement costs, fraud losses, as well as the removal of malicious and accidental exceptions. 

Ethereum supports two types of accounts [2]:An externally owned account (user-controlled) is controlled by a user. We denote these accounts by EOA.A contract account is controlled by the smart contract itself. We denote a contract account by CA.

Both account types can store the Ethereum currency ‘ether’. Ethereum does not perform functions (computations) in a smart contract without user interaction. Thus, a CA must be activated by an EOA before its functions are executed. Executing functions requires the EOA to purchase ‘gas’ using the ether currency.

‘Gas price’ determines the conversion rate of gas to ether. The ‘gas’ is essentially a transaction fee to encourage miners to include transaction executions into blocks of the public Ethereum blockchain. As such, gas is a standardizing metric that estimates the cost of executing code on the Ethereum network. Each assembly operation (opcode) has a fixed gas cost based on its expected execution time. 

‘Gas limit’ is set to prevent infinite loops, which would abuse the resources on the Ethereum block. If the limit is exceeded, the transactions are not completed, and the corresponding block is not mined. 

The structure of an Ethereum transaction is illustrated in Table 1:From: a signature from the owner of the EOA is needed to authorize the transaction.To: the receiver of the transaction can either be an EOA or CA.Data: the code to deploy a new contract or to execute transactions for the contract.Gas price: the conversion rate from gas to ether currency.Total gas: the maximum amount of gas that may be consumed by the transaction.Nonce: a counter that is incremented per each new transaction from an account.

The Ethereum blockchain can be considered an orderly transaction-based state machine. If multiple transactions invoke the same contract, then the final state is determined by the order of transactions that are stored in the block.

Blockchain relies upon *miners* performing ‘proof of work’, which entitles the chosen miner to append a new block. This proof of work is a computationally intensive race to find a nonce, which results in a hash value below the preset threshold. The successful miner is rewarded five ethers. 

Ethereum blockchain is also selected for the Open Vote Network [1]. 

## 3. Technical Bottlenecks in Ethereum Implementation and Proposed Solutions

We identified three major bottlenecks in implementing the above voting protocol on the Ethereum blockchain: (1) division problem in encryption of voting values; (2) large time complexity in tallying; and (3) tallying failure due to “no votes” from registered voters. 

We describe the bottlenecks and propose our solutions. The solutions are implemented in our scalable and anonymous voting system on Ethereum.

### 3.1. Division Problem in Encryption of Voting Values

Voter *i* publishes $g^{x_{i}y_{i}}g^{v_{i}}$ as an encrypted voting value to guarantee anonymity. The *v_i_* represents whom voter *i* votes for and, thus, should not be revealed. For a yes/no vote in a single candidate case [1], *v_i_* is one of {1,0}. For multiple candidates, *v_i_* can be coded to facilitate tallying as in Equation (12) [7]. For candidate *j*, *v_i_* becomes $2^{jm}$. In our scheme, if voter *i* votes for candidate *j*, then *v_i_* = *t_j_*, where *t_j_* is the *j*-th smallest prime number as in Equation (14).
(14)$$t_{0}=2,\;t_{1}=3,\;t_{2}=5\ldots$$

Equation (14) simply changes the encoding in Equation (12) to unique prime numbers, which also guarantees unambiguous resolution and further facilitates recovery from “no vote after registration”

To compute $g^{\;y_{i}}$ in Equation (7), voter i(i≠j) collects $g^{x_{j}}$ for all j(j≠i,j=0 to n−1) and performs the division [1,6]. While $x_{j}$ is hidden as a private key of node *j*, the $g^{x_{j}}$ is broadcast as its corresponding public key. The division introduces a round-off error, which makes it hard to obtain the exact value. Figure 1 illustrates this problem with an example (*p* = 1091659, *q* = 181943, *g* = 9, $x_{1}$ = 5, $x_{2}$ = 2, $x_{3}$ = 3). The answer should be 1 instead of 0.9999999997, as shown in the screen printout. 

Modular division using the cyclic group [16] can be used to prevent this. The modular division is performed by a multiplication by its inverse. Computing the multiplicative inverse in the modular computation takes *O*((log *p*)^2^) time using the extended Euclidean algorithm [17].

#### Our Solution 

To circumvent this, we replace divisions by multiplications exploiting a feature of the cyclic group. Let *p* and *q* be prime numbers where (p−1) mod q=0 and${\;g}^{q}\;\mathrm{mod}\;p=1$. Then, we have $g^{-x}\;\mathrm{mod}\;p=g^{\;q-x}\;\mathrm{mod}\;p$ from the cyclic group theory [29]. This implies a division by $g^{x}$ in this cyclic group can be replaced by a multiplication by $g^{q-x}$.

However, $x_{i}$ is hidden as a private key of node *j*, thus voter i(i≠j) cannot possibly compute ($q-x_{i}$) from $g^{x_{i}}$. If node i only receives $g^{x_{j}}\;$as in a previous scheme [1], it is unable to compute $g^{q-x_{j}}$. To solve this problem, we arrange for node j(j=0 to n−1) to precompute $g^{q-x_{j}}$ and broadcasts it along with $g^{x_{j}}$. 

Although voter i broadcasts $g^{q-x_{i}}$ along with $g^{x_{i}}$, any receiver is unable to compute $q-x_{i}$ and $x_{i}$ with $g^{q-x_{i}}$ and $g^{x_{i}}$ [21]. Even if a malicious attacker k gets $x_{k}$
$\left(g^{x_{k}}\mathrm{mod}\;p=g^{x_{i}}\mathrm{mod}\;p\right)$, voter i still can protect his own key $x_{i}$ because $x_{k}=nq+x_{i}$
(n=0,1,2,3,…) in the cyclic group [29].

Now, we can replace the division in Equation (15) by a multiplication, as shown in Equations (16) and (17).
(15)$$Y_{i}=g^{y_{i}}=\prod_{j=0}^{i-1}g^{x_{j}}/\prod_{j=i+1}^{n-1}g^{x_{j}}$$
(16)$$g^{y_{i}}=\prod_{j=0}^{i-1}g^{x_{j}}\times \prod_{j=i+1}^{n-1}g^{q-x_{j}}$$
(17)$$\begin{array}{ll} y_{i} & =\sum_{j=0}^{i-1}x_{j}+\sum_{j=i+1}^{n-1}\left(q-x_{j}\right)\\ & =\sum_{j=0}^{i-1}x_{j}-\sum_{j=i+1}^{n-1}x_{j}+(n-i-1)q \end{array}$$

This solves the division problem in *O*(1) time instead of *O*(log(*p*)^2^) time using the extended Euclidean algorithm.

### 3.2. Large Time Complexity in Tallying

For *i* = 0, 1, 2, …, *n* − 1, voter *i* broadcasts $g^{x_{i}y_{i}}g^{v_{i}}$ to all other voters. After collecting all broadcasts, each voter performs verification of arriving NIZKP proofs first, and then computes Equation (9) for tallying on its own. By Proposition 1, in the above, each voter executes Equation (18) to obtain a tally value Z. Each voter resolves this tally value to identify how many votes each candidate has won.
(18)$$\prod_{i=0}^{n-1}g^{x_{i}y_{i}}g^{v_{i}}=\prod_{i=0}^{n-1}g^{v_{i}}=g^{\sum_{i=0}^{n-1}v_{i}}=Z$$

First, we consider a previous tallying scheme for multiple candidates by Baudron et al. [7]. The encoding scheme, to represent a vote for each candidate, is of super-increasing nature as dictated in Equations (11)–(13). We illustrate the scheme in an example where each of the four voters vote for one of the four candidates (*n* = 4, *k* = 4). We assume g = 3 for simplicity and all voters share a tally value = $g^{\sum v_{i}}=3^{137}$ and try to tally individually.
(19)$$g^{137}=g^{\sum_{i=0}^{3}v_{i}}=g^{2^{0\times 3}\cdot c_{0}+2^{1\times 3}\cdot c_{1}+2^{2\times 3}\cdot c_{2}+2^{3\times 3}\cdot c_{3}}$$

Then the tallying in this example is to obtain a unique 4-tuple <$c_{0}{,c}_{1}{,c}_{2}{,c}_{3}$> from Equation (19). The <$c_{0}{,c}_{1}{,c}_{2}{,c}_{3}$> is a 4-tuple representing the numbers of votes won by candidates 0, 1, 2, and 3, respectively. 

In their scheme, all possible 4-tuples are pre-computed for a mapping table, which pairs each 4-tuple with the corresponding tally value. Once the tally value is obtained, the table can be used to identify the corresponding 4-tuple. The number of possible 4-tuples is _4_*H*_4_ = _4+4−1_*C*_3_.

In general, if the number of candidates and the number of voters are *k* and *n*, respectively, the mapping table would have *_k_H_n_* = *_n_*_+*k*−1_*C_k_*_−1_ entries, which can grow very large, even for moderate values of *n* and *k*. Thus the construction of the table, or to identify the *k*-tuple ($c_{0}{,c}_{1}{,c}_{2}{,\text{…},c}_{k-1}$) for *n* voters, may take *O*(*n^k^*) time. Therefore, this method can be used only for small values of *n, k*, severely limiting the scalability of this voting scheme. Otherwise, the number of possible results easily explode beyond practical implementation (the gas needed for program execution can easily exceed the gas limit).

#### Our Solution

First, we devised a new encryption scheme for voting values. In the previous scheme by Hao, Ryan, and Zielinski [6], voter *i* broadcasts $g^{x_{i}y_{i}}g^{v_{i}}\;\mathrm{mod}\;p$ as its encrypted voting value. Thus, a tally value is $\underset{i}{\prod }g^{x_{i}y_{i}}g^{v_{i}}{\;\mathrm{mod}\;p=g}^{\underset{i}{\sum }v_{i}}\;\mathrm{mod}\;p$. 

Our new encryption scheme for voting value for voter *i* is as in Equation (20).
(20)$$v_{i}\times g^{x_{i}y_{i}}\mathrm{mod}p$$

In our scheme, $v_{i}$ is a prime number, which represents a candidate. For example, we have three candidates, which are represented by three prime numbers, 2, 3, and 5, respectively. An attacker may try to reveal $v_{i}\;$by dividing the encrypted voting value $v_{i}\times g^{x_{i}y_{i}}\mathrm{mod}p$ with 2, 3, or 5 to see if the remainder is zero. However, this attack does not reveal $v_{i}$ since $g^{x_{i}y_{i}}\mathrm{mod}p$ belongs to [0, *p*−1] and there exist many multiples of 2, 3, 5, 2 × 3, 2 × 5, 3 × 5 and 2 × 3 × 5 in [0, *p*−1] for large *p*. If $v_{i}\times g^{x_{i}y_{i}}\mathrm{mod}p$ is a multiple of 30, the attacker cannot tell if $v_{i}\;$is 2, 3, or 5. 

One may find our scheme weak in the anonymity requirement. “If $v_{i}\times g^{x_{i}y_{i}}$ is a multiple of 30, the attacker cannot tell if $v_{i}$ is 2, 3, or 5.” That is true, but he can say it is not 7 or 11 and so on”. We rebut this claim with a counter example where *p* = 11, *q* = 5, *g* = 3, $v_{i}$ = 3 and $g^{x_{i}y_{i}}$ = 9. $v_{i}\times g^{x_{i}y_{i}}\mathrm{mod}p$ = 5 and it is not divisible by 3, but there still exists 3 in $v_{i}\times g^{x_{i}y_{i}}$. 

Second, our encoding scheme to represent a vote for each candidate is different from Baudron et al. [7] (refer to Equations (11)–(13)).

In our scheme, if voter *i* votes for candidate *j*, then *v**_i_* = *t**_j_*, where *t**_j_* is the *j*-th smallest prime number as in Equation (14).

Thus, the tally value in our scheme is as in Equations (21) and (22), using Proposition 1.
(21)$$\begin{array}{c} \prod_{i=0}^{n-1}v_{i}g^{x_{i}y_{i}}\mathrm{mod}p=\prod_{i=0}^{n-1}v_{i}\mathrm{mod}p=\prod_{i=0}^{n-1}v_{i}=Z\\ \;(\mathrm{assuming}\;\prod_{i=0}^{n-1}v_{i}<p) \end{array}$$
(22)$$Z=t_{0}{}^{c_{0}}\times t_{1}{}^{c_{1}}\times \ldots .\times t_{k-1}{}^{c_{k-1}}$$

The tallying is to identify the k-tuple ($c_{0},c_{1},c_{2},\ldots ,c_{k-1}$) in Equation (22). We illustrate our scheme in Equation (23) as an example where each of the four voters vote for one of the four candidates (n=4,k=4). We assume g=3 and all voters share a tally value Z = 126.
(23)$$\begin{array}{l} Z=126\\ =t_{0}^{c_{0}}\times t_{1}^{c_{1}}\times t_{2}^{c_{2}}\times t_{3}^{c_{3}}\\ {=2}^{c_{0}}\times 3^{c_{1}}\times 5^{c_{2}}\times 7^{c_{3}}\\ {=2}^{1}\times 63\;(c_{0}=1)\\ {=2}^{1}\times 3^{2}\times 7\;(c_{1}=2)\\ {=2}^{1}\times 3^{2}\times 5^{0}\times 7\;(c_{2}=0)\\ {=2}^{1}\times 3^{2}\times 5^{0}\times 7\;(c_{3}=1) \end{array}$$

For *j* = 0 to *k* − 1, *c**_j_* = [0, *n*]. Note that *c**_j_* is the maximum *m**_j_* such that Z mod $t_{j}^{m_{j}}$ is equal to 0. Since *c**_j_* can be identified in *O*(log *n*) steps, we have time complexity of *O*(*k*log *n*), which is asymptotically smaller than the *O*(*n^k^*) time in Baudron et al. [7]. 

### 3.3. Tallying Failure Due to “No Votes” from Registered Voters

We identify this problem from previous voting schemes [6,7]. The tallying is not possible if one or more registered voters fail to send their encrypted voting values. This is because Equation (12), which is used to retrieve the encrypted tally value, depends on Proposition 1, which requires that all registered voters send their encrypted voting values. In reality, we may have this failure more often than not.

If we have four voters and four candidates, then we expect all four voters to send their encrypted voting values for the tally value to be retrieved, as in Equation (24).
(24)$$\prod_{i=0}^{3}v_{i}g^{x_{i}y_{i}}\mathrm{mod}p=\prod_{i=0}^{3}v_{i}$$

If we assume voter 3 did not send his/her voting value, then Equation (24) changes to Equation (25).
(25)$$\begin{array}{l} \prod_{i=0}^{2}v_{i}g^{x_{i}y_{i}}\mathrm{mod}p=\prod_{i=0}^{2}v_{i}\cdot g^{\sum_{i=0}^{2}x_{i}y_{i}}\mathrm{mod}p\\ =\prod_{i=0}^{2}v_{i}\cdot g^{x_{0}(-x_{1}-x_{2}-x_{3})+x_{1}(x_{0}-x_{2}-x_{3})+x_{2}(x_{0}+x_{1}-x_{3})}\mathrm{mod}p\\ =\prod_{i=0}^{2}v_{i}\cdot g^{-x_{0}x_{3}-x_{1}x_{3}-x_{2}x_{3}}\mathrm{mod}p\\ =\prod_{i=0}^{2}v_{i}\cdot g^{-x_{3}y_{3}}\mathrm{mod}p \end{array}$$

Certainly we could not resolve the incomplete tally value in Equation (25) into the correct 4-tuple <$c_{0}{,c}_{1}{,c}_{2}{,c}_{3}$>.

#### Our Solution

Our solution is to remove the $g^{-x_{3}y_{3}}$ in the tally value at the end of Equation (25) by a multiplication with $g^{x_{3}y_{3}}$. We show how to compute $g^{x_{3}y_{3}}$ without help from voter 3, who fails to send his/her encrypted voting value.

After the voting round expires, every voter is notified about which voter(s) has (have) failed to vote. First, we show how to recover from the failure of a single voter and then extend our algorithm to multiple failed voters. 

Assume a registered voter *j* out of *n* voters has failed to vote. Upon discovering this, voter *i* broadcasts an additional value, according to Equation (26). Voter *i* knows its private key $x_{i}$, but does not know $x_{j}$. However, it knows $g^{\;x_{j}}$ when voter *j* has registered.
(26)$$\left\{ \begin{array}{ll} g^{x_{i}x_{j}}\mathrm{mod}p & if\;i<j\\ g^{(q-x_{i})x_{j}}\mathrm{mod}p & if\;i>j \end{array}\right.$$

By multiplying all the values broadcast by all the voters except *j*, each voter can obtain $g^{x_{j}y_{j}}$ as shown in Equation (27).
(27)$$\begin{array}{l} \prod_{i=0}^{j-1}g^{x_{i}x_{j}}\mathrm{mod}p\times \prod_{i=j+1}^{n-1}g^{(q-x_{i})x_{j}}\mathrm{mod}p\\ =g^{x_{0}x_{j}+x_{1}x_{j}+\ldots +x_{j-1}x_{j}-x_{j+1}x_{j}-\ldots -x_{n-1}x_{j}+(n-j-1)q}\mathrm{mod}p\\ =g^{x_{j}y_{j}}\mathrm{mod}p \end{array}$$

This solves the failure of a single voter, as shown in Equation (28)
(28)$$\begin{array}{l} (\prod_{i=0}^{2}v_{i}\cdot g^{-x_{3}y_{3}}\mathrm{mod}p)\times (g^{x_{0}x_{3}}\cdot g^{x_{1}x_{3}}\cdot g^{x_{2}x_{3}})\mathrm{mod}p\\ =\prod_{i=0}^{2}v_{i}\cdot g^{-x_{3}(x_{0}+x_{1}+x_{2})}g^{x_{3}(x_{0}+x_{1}+x_{2})}\mathrm{mod}p\\ =\prod_{i=0}^{2}v_{i}\cdot g^{-x_{3}(x_{0}+x_{1}+x_{2})+x_{3}(x_{0}+x_{1}+x_{2})}\mathrm{mod}p\\ =\prod_{i=0}^{2}v_{i}\mathrm{mod}p\\ =v_{0}\cdot v_{1}\cdot v_{2} \end{array}$$

Now, we extend our scheme with a case of two failed voters, 2 and 3 out of 0, 1, 2, and 3. The incomplete tally value would be like Equation (29).
(29)$$\begin{array}{l} \prod_{i=0}^{1}v_{i}g^{x_{i}y_{i}}\mathrm{mod}p=v_{0}v_{1}g^{\sum_{i=0}^{1}x_{i}y_{i}}\mathrm{mod}p\\ =v_{0}v_{1}g^{x_{0}(-x_{1}-x_{2}-x_{3})+x_{1}(x_{0}-x_{2}-x_{3})}\mathrm{mod}p\\ =v_{0}v_{1}g^{-x_{0}x_{2}-x_{0}x_{3}-x_{1}x_{2}-x_{1}x_{3}}\mathrm{mod}p \end{array}$$

On finding the failure of voters 2 and 3, voter 0 and voter 1 send two additional values each, respectively, according to Equation (30).
(30)$$\{\begin{array}{cc} g^{x_{0}x_{2}}\mathrm{mod}p,\;g^{x_{0}x_{3}}\mathrm{mod}p & if\;i=0\\ g^{x_{1}x_{2}}\mathrm{mod}p,\;g^{x_{1}x_{3}}\mathrm{mod}p & if\;i=1 \end{array}$$

Then voter 0 and voter 1 can retrieve the correct tally value as in Equation (31).
(31)$$\begin{array}{l} v_{0}\cdot v_{1}\cdot g^{-x_{0}x_{2}-x_{0}x_{3}-x_{1}x_{2}-x_{1}x_{3}}\mathrm{mod}p\times \\ (g^{x_{0}x_{2}}\cdot g^{x_{0}x_{3}}\cdot g^{x_{1}x_{2}}\cdot g^{x_{1}x_{3}})\mathrm{mod}p\\ =v_{0}\cdot v_{1}\cdot g^{-x_{0}x_{2}-x_{0}x_{3}-x_{1}x_{2}-x_{1}x_{3}+x_{0}x_{2}+x_{0}x_{3}+x_{1}x_{2}+x_{1}x_{3}}\mathrm{mod}p\\ =v_{0}\cdot v_{1}\cdot \mathrm{mod}p \end{array}$$

Now we extend this to the general case in which *m* out of *n* voters have failed to vote. Let *U* denote a set of *m* failed voters. We have $U=\left\{u_{0},u_{1},u_{2},\ldots ,u_{m-1}\right\}$ and $\left(0\leq u_{0}<u_{1}<\ldots <u_{m-1}\leq n-1\right)$. 

Then, the incomplete tally value would be like in Equation (32).
(32)$$\begin{array}{l} \prod_{i=0(i\notin U)}^{n-1}v_{i}g^{x_{i}y_{i}}\mathrm{mod}p=\prod_{i=0(i\notin U)}^{n-1}v_{i}\cdot g^{\left(\sum_{i=0(i\notin U)}^{n-1}x_{i}y_{i}\right)}\mathrm{mod}p\\ =\prod_{i=0(i\notin U)}^{n-1}v_{i}\cdot g^{\left(-\sum_{i=0}^{n-1}\sum_{u_{j}=u_{0}}^{u_{j}<i}x_{i}x_{u_{j}}+\sum_{i=0}^{n-1}\sum_{u_{j}>i}^{u_{m-1}}x_{i}x_{u_{j}}\right)}\mathrm{mod}p \end{array}$$

Then voter *i* who does not belong to *U* should send additional values according to Equation (33).
(33)$$g^{\left(x_{i}\sum_{u_{j}=u_{0}}^{u_{j}<i}x_{u_{j}}-x_{i}\sum_{u_{j}>i}^{u_{m-1}}x_{u_{j}}\right)}\mathrm{mod}p$$

Multiplying all of the additional values broadcast by the *n* − *m* voters who do not belong to *U*, each voter is able to obtain the value in Equation (34).
(34)$$\begin{array}{l} \prod_{i=0(i\notin )}^{n-1}g^{x_{i}\sum_{u_{j}=u_{0}}^{u_{j}<i}x_{u_{j}}-x_{i}\sum_{u_{j}>i}^{u_{m-1}}x_{u_{j}}}\mathrm{mod}p\\ =g^{\sum_{i=0}^{n-1}\sum_{u_{j}=u_{0}}^{u_{j}<i}x_{i}x_{u_{j}}-\sum_{i=0}^{n-1}\sum_{u_{j}>i}^{u_{m-1}}x_{i}x_{u_{j}}}\mathrm{mod}p \end{array}$$

Now, the correct tally value can be retrieved by multiplying Equation (32) by Equation (34) to obtain Equation (35).
(35)$$\begin{array}{c} (\prod_{i=0(i\notin U)}^{n-1}v_{i}\cdot g^{-\sum_{i=0}^{n-1}\sum_{u_{j}=u_{0}}^{u_{j}<i}x_{i}x_{u_{j}}+\sum_{i=0}^{n-1}\sum_{u_{j}>i}^{u_{m-1}}x_{i}x_{u_{j}}}\cdot g^{\sum_{i=0}^{n-1}\sum_{u_{j}=u_{0}}^{u_{j}<i}x_{i}x_{u_{j}}-\sum_{i=0}^{n-1}\sum_{u_{j}>i}^{u_{m-1}}x_{i}x_{u_{j}}})\mathrm{mod}p\\ =(\prod_{i=0(i\notin U)}^{n-1}v_{i}\cdot g^{-\sum_{i=0}^{n-1}\sum_{u_{j}=u_{0}}^{u_{j}<i}x_{i}x_{u_{j}}+\sum_{i=0}^{n-1}\sum_{u_{j}>i}^{u_{m-1}}x_{i}x_{u_{j}}+\sum_{i=0}^{n-1}\sum_{u_{j}=u_{0}}^{u_{j}<i}x_{i}x_{u_{j}}-\sum_{i=0}^{n-1}\sum_{u_{j}>i}^{u_{m-1}}x_{i}x_{u_{j}}})\mathrm{mod}p\\ =(\prod_{i=0(i\notin U)}^{n-1}v_{i})\mathrm{mod}p \end{array}$$

This solves the general case of *m* (0 *< m < n* − 1) failed voters out of *n*.

## 4. Performance Evaluation

We use Truffle [30] and Remix [31] running on a desktop PC with four cores (3.40GHz Intel Core i5-7500 and 64 GB DDR4 RAM) for performance evaluation. A private Ethereum is built with Truffle and transactions are generated by using Remix [31]. The simulation is performed with different pairs of (*n*,*k*) where *n*, *k* are the number of voters and the number of candidates, respectively. The performance of our scheme is evaluated in terms of scalability, *gas* cost, and time complexity.

### 4.1. Scalability

We define scalability in e-voting system as the maximum number of allowed voters and/or candidates for given values of *p*.

Figure 2 compares the number of allowed candidates when there are 60 voters for our system against McCorry, Shahandahti, and Hao [1] and Hao, Ryan, and Zieliński [6]. 

Up to two candidates (or yes/no vote for one candidate) are allowed in [1]. Up to 60 candidates are allowed in [6], but only the protocol is provided without blockchain implementation. 

As shown in Equation (21), the number of allowed candidates may increase as *p* increases before the gas limit is reached in our scheme, allowing for scalability in terms of candidates. For instance, the number of allowed candidates is around 1000 for *p* > $10^{230}$.

Figure 3 compares the maximum number of allowed voters when there are two candidates for our scheme against McCorry, Shahandahti, and Hao [1] and Hao, Ryan, and Zieliński [6]. Up to 60 voters are allowed in [1] and 112 voters are allowed in [6]. Since only the protocol is provided in [6], we implement the protocol on our own for the purpose of comparison. 

In our system, the maximum number of voters may increase as *p* increases, as shown in Figure 3. This shows the scalability of our scheme in terms of voters. For instance, 1000 voters are allowed for *p* > $10^{300}$. 

Figure 4 compares the maximum number of allowed voters when there are 10 candidates for our scheme against Hao, Ryan, and Zieliński [6]. While six voters are allowed in [6], our scheme allows more voters as *p* increases, as shown in Figure 4.

In terms of scalability, our scheme can hold large-scale voting by using a huge value of *p*. For example, our scheme can hold a voting in a town, which has 5000 voters and 10 candidates, by choosing $p\approx 10^{5000}$.

### 4.2. Gas Cost

Gas is a unit that measures the amount of computational effort required to execute specific transactions on the Ethereum blockchain [8].

Gas fees keep the Ethereum blockchain network from abuse by malicious or irresponsible nodes. Requiring fees for computations to be executed on the network prevents abusers from spamming the network. To prevent possible infinite loops, each transaction is asked to set a limit on how many computational steps to use. The fundamental unit of computation is *gas* and the limit for specific transaction is called *gas limit*. The transaction is aborted if the preset gas limit is reached before finish.

In this section, we compare gas costs for our scheme against previous schemes [1,6]. 

Figure 5 compares the gas cost of the tallying transaction in our scheme against McCorry, Shahandahti, and Hao [1] and Hao, Ryan, and Zieliński [6] when the number of candidates (*k*) is 2. 

As the number of voters (*n*) grows, the gas costs for tallying increases in [1] and [6] while the gas cost of our scheme grows very little. If we preset the gas limit as $3*10^{6}$ (red dotted line), the tallying transaction for 40 voters for [1] may cost more than the limit and, hence, will be aborted. 

For 60 voters, our scheme costs around $0.09*10^{6}$ gas, while the scheme in [1] costs $4.7*10^{6}$ gas. Our scheme excels [6] by reducing gas cost to 1/53 of the scheme in [6], showing the scalability of our scheme. 

Figure 6 shows gas costs vs. number of candidates (*k*) in our scheme when the number of voters (*n*) is 10, 30, 50, 100, and 200. It shows the gas cost increases modestly as n and/or k increases. At the maximum (*n* = 200, *k* = 50), the gas cost is slightly over $1*10^{6}$.

Figure 5 and Figure 6 plots gas costs against the number of voters and candidates in our simulation with limited pairs of (candidates, voters) and a chosen *p*. The *p* should grow to accommodate a large number of voters or candidates. Simulation time would grow too large to be finished in a reasonable amount of time. Since gas cost is proportional to the amount of computations, which grows in *O*(*k*log *n*) where *k*, *n* denote the number of candidates and the number of voters, respectively. We expect the total gas would grow similarly. However, we do not claim our scheme can be directly applied to “big voting”, such as a national-scale vote in its current form. Rather, we claim our scheme extends a previous small-scale blockchain-based voting system to a medium-size one.

### 4.3. Computational Burden for Tallying

Figure 7 shows how fast the number of possible tally combinations increases for Hao, Ryan, and Zieliński [6], which uses the protocol in Baudron et al. [7]. If a mapping table is to be constructed for all possible tallying combinations as entries, the size of the table may grow formidably as *n* and *k* grow. For instance, the number of entries grows up to $10^{19}$ for 10 candidates and 500 voters, greatly increasing the time to construct or search the corresponding table.

Figure 8 shows the number of computations needed for tallying in our scheme as the number of candidates (*k*) grows. Note that the numbers in our scheme are much smaller than those in Figure 7. 

This implies our scheme greatly reduces the number of computations. For 10 candidates and 500 voters, our scheme requires 1/$10^{17}$ of the computations needed for tallying in Hao, Ryan, and Zieliński [6] and Baudron et al. [6,7], as shown in Figure 7. 

Figure 9 compares our scheme against Hao, Ryan, and Zieliński [6] and Baudron et al. [7]. The number of computations in our scheme grows much slower than that of [6], exhibiting the scalability of our scheme when implemented on Ethereum blockchain.

## 5. Conclusions

We identified three major bottlenecks in implementing anonymous and scalable voting on the Ethereum blockchain. These include the division problem in the encryption of voting values, large time complexity in tallying, and tallying failure due to “no votes” from registered voters. Algorithmic solutions and implementation details are provided. An experiment using Truffle and Remix running on a desktop PC was performed for evaluation. Our scheme shows great reduction in gas needed for execution of anonymous voting on Ethereum.

For instance, our scheme consumes 1/53 of the gas needed for a state-of-the-art solution for 60 voters. Time complexity analysis shows our scheme is asymptotically superior to known solutions. Finally, we proved our solution in regards to the tallying failure due to “no votes” from registered voters. To the best of our knowledge, our solution is the first to address this problem in the literature.

## Figures and Tables

**Figure 1 sensors-21-03958-f001:**
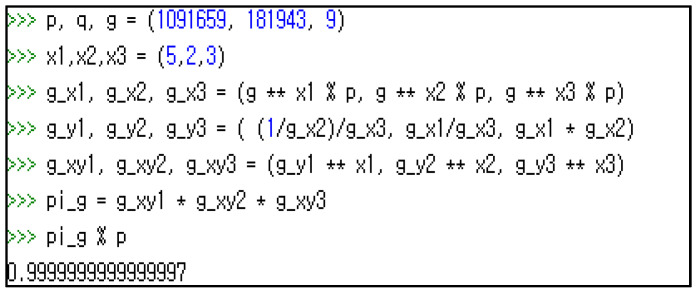
Division error in computing $\prod g^{x_{i}y_{i}}\mathrm{mod}p$.

**Figure 2 sensors-21-03958-f002:**
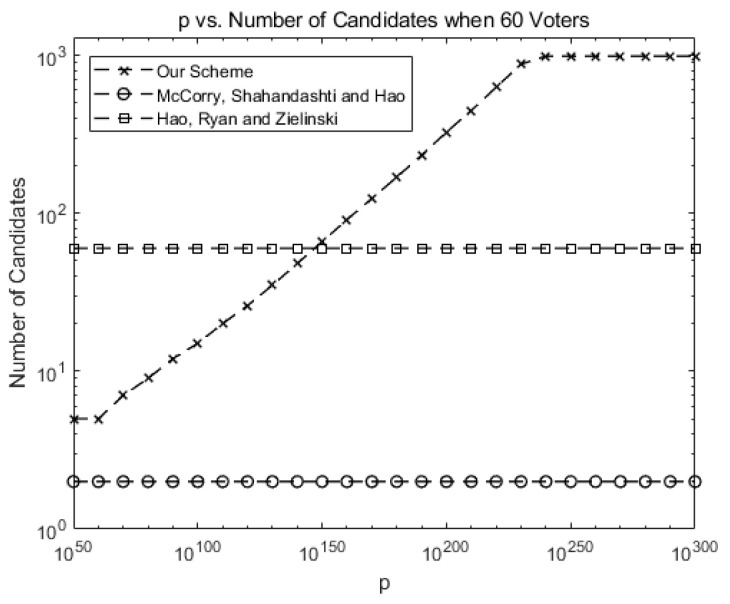
Comparison: the number of allowed candidates for our scheme against previous schemes (number of voters *n* = 60).

**Figure 3 sensors-21-03958-f003:**
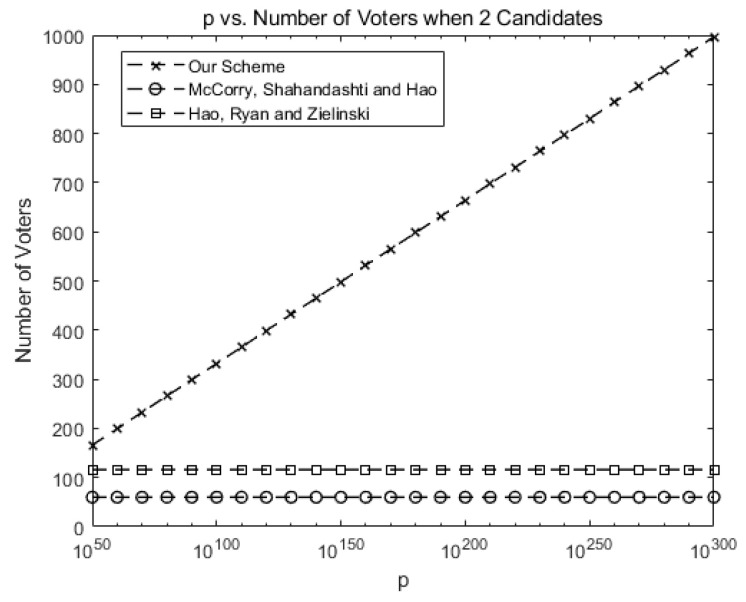
Comparison of the number of allowed voters when there are two candidates for our scheme against previous schemes (number of candidates *k* = 2).

**Figure 4 sensors-21-03958-f004:**
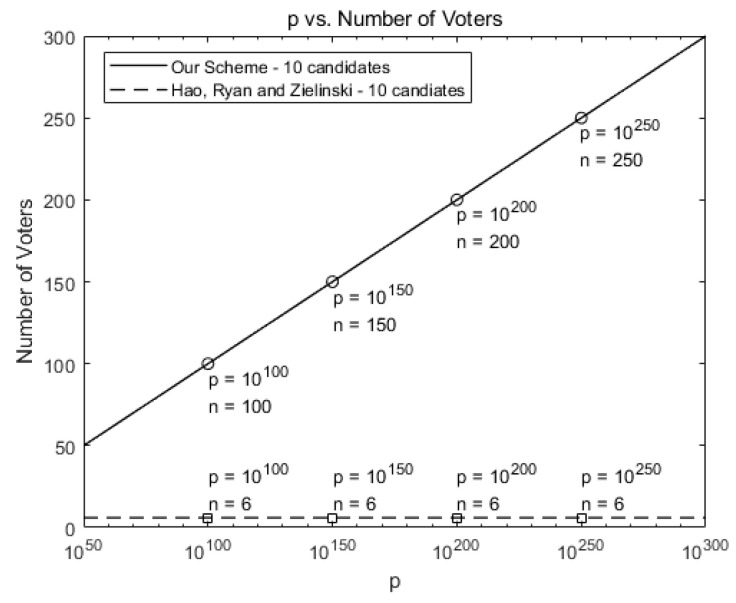
Comparison of the maximum number of allowed voters for our scheme against [6] (number of candidates *k* = 10).

**Figure 5 sensors-21-03958-f005:**
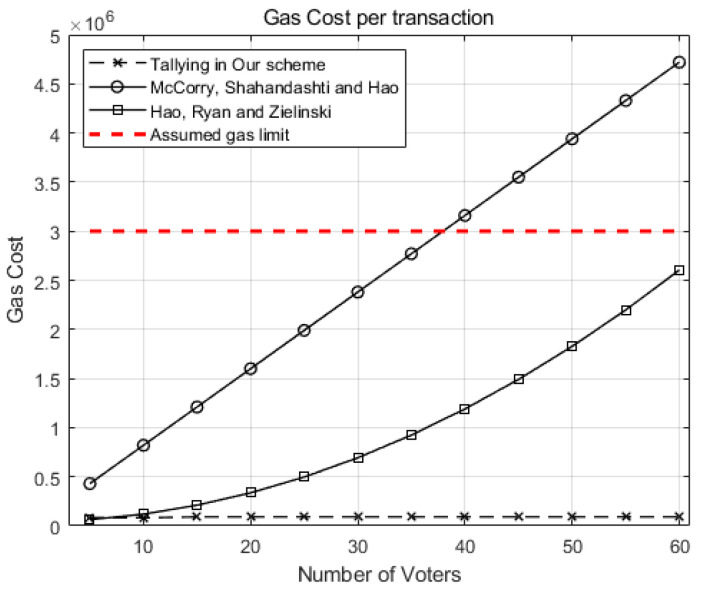
Comparison of the gas costs for a tallying transaction in our scheme against McCorry, Shahandahti, and Hao [1] and Hao, Ryan, and Zieliński [6].

**Figure 6 sensors-21-03958-f006:**
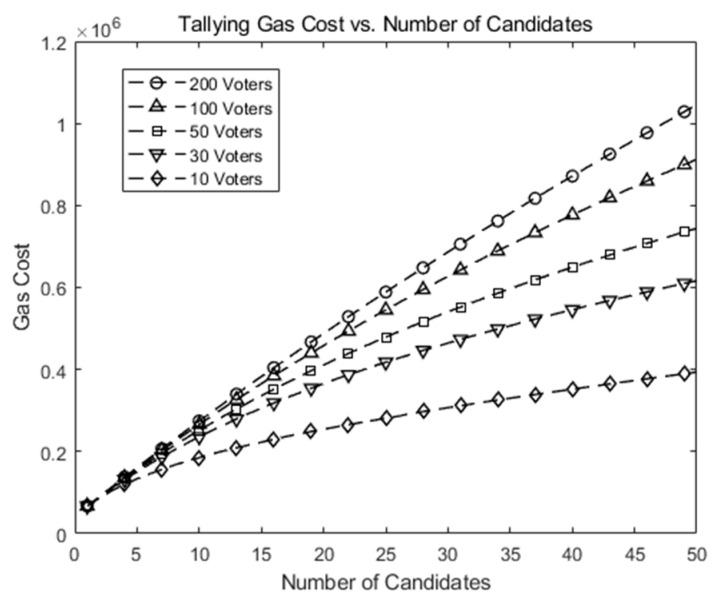
Gas cost for a tallying transaction in our scheme as the number of candidates (*k*) increases.

**Figure 7 sensors-21-03958-f007:**
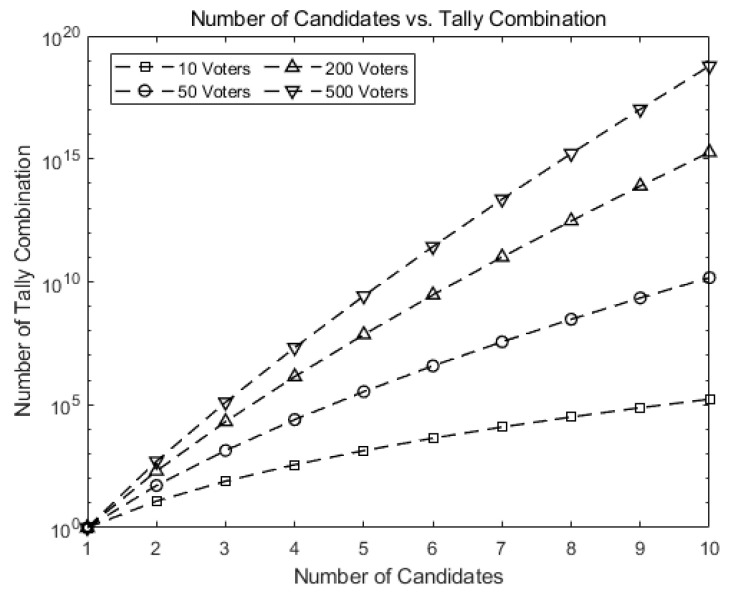
Number of possible tally combinations increases for Hao, Ryan, and Zieliński [6], and Baudron et al. [7].

**Figure 8 sensors-21-03958-f008:**
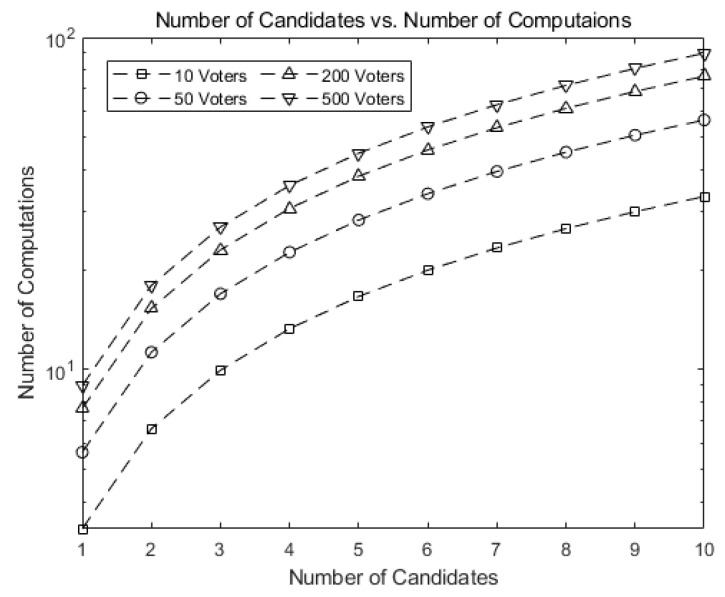
Number of computations for tallying in our scheme as the number of candidates (*k*) grows.

**Figure 9 sensors-21-03958-f009:**
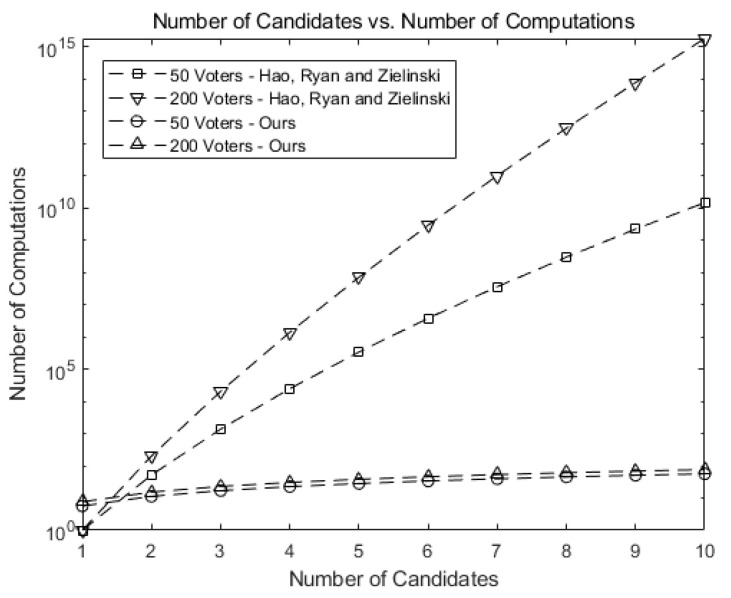
Comparison of the number of computations for our scheme against [6].

**Table 1 sensors-21-03958-t001:** Illustration of the structure of an Ethereum transaction.

Field	Example
From	EOA (External Owned Account) A
To	CA (Contract Address)
Gas Price	$2\times 10^{-8}$ ether
Total Gas	2,000,000
Nonce	35
Data	Contract code made by A

## Data Availability

Not applicable.

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
