# Peer review of "A Scalable Implementation of Anonymous Voting over Ethereum Blockchain"

_sensors, 2021, doi:10.3390/s21123958_

Round 1

Reviewer 1 Report

The authors have identified three major bottlenecks in implementing anonymous and scalable voting on Ethereum blockchain.  This paper is readable and well-written.

However, I also have some comments on this paper. This paper appears to have been written quite some time ago and is only now being put forward for publication. The majority of the references are from the 2017's. There are several works that have been published in recent years. The fact that these are not contained in the literature review, leads me to question whether the research presented is current enough to be published. In order for the manuscript to be publishable, the author(s) will have to clearly demonstrate how they contribute to the current state of research in the area. 

List some suggestions:

  1. Investigating performance constraints for blockchain based secure e-voting system KM Khan, J Arshad, MM Khan - Future Generation Computer Systems, 2020
  2. Efficient, coercion-free and universally verifiable blockchain-based voting T Dimitriou - Computer Networks, 2020 - Elsevier
  3. Allowing non-identifying information disclosure in citizen opinion evaluation F Buccafurri, L Fotia, G Lax - … on Electronic Government and the Information …, 2013

For security analysis, the authors must list several attack cases in real world and analyze the security of our model based on these assumptions. I keep in my mind that the formal security proof should be given to prove the security for the proposed protocol.

Reviewer 2 Report

This paper examines two previous works about electronic voting, showing flaws and proposing feasible alternatives in the protocol. The underlying idea is to use a modified Zero-Knowledge-Proof protocol on an Ethereum blockchain.
The paper is interesting, as it spots severe weakness in the ZKP voting protocols adopted, but the authors missed to prove that their modification does not prejudice the security of the chosen protocols. For example, they substitute eq. (12) with eq. (14), without providing any algebraic proof of equivalence, which is essential to ensure the anonymity of the "ballot." A similar problem happens with the introduction of eq. (20), which significantly changes the quantity on line 177, and the authors do not even try to present a proof of equivalence.
The authors do not convince that the voter's private key remains private when broadcasting b^(q-x_j) along with b^(x_j). Proof should be provided.
About performances, the presentation shows that the costs and computational complexity of this implementation vs. the other solutions examined is significantly better. The point is that the numbers presented are not even close to the numbers of actual voting in a little-medium-sized town. Therefore, the authors should point out the target of their protocol and afford the aspects relative to real issues present in their target. The honestly cited paper [10] presents issues that should be examined in every paper about electronic or blockchain voting protocol, and I think the authors should address the critics about ZKP.
Finally, the authors should scrutinize errors in symbols (e.g., line 148 with wrong letter size; missing definitions on lines 153, 158; unreadable stress mark in Eq. (11)) and English syntax. An inconsistent paragraph from line 260 to 267 must be removed.

Round 2

Reviewer 2 Report

The authors improved the text by correcting some formulas and adding details on critical novelties introduced in their new model.

The authors still do not convince on conclusions in some statements. Moreover, the new version does not contain all the changes they reported in the "author response" file, and some statements are significantly different.

In details:

  • "If v_i x g^{x_i y_i} is a multiple of 30, the attacker can not tell if v_i is 2, 3, or 5." That is true, but he can say it is not 7 or 11 and so on. This property is a severe weakness on anonymity requirements. The reader can imagine applying combinatorics that, in some cases, can expose voting intentions. 
  • Lines 312-317: the text on the newly submitted version is different from the text on the "author response." In some way, the new text is better than the commented one, but there is a statement not clear: "each value is unable to compute − x_i and x_i from any computation with g^{qx_i} and g^{x_i} respectively." How can a value compute anything? Maybe there is a grammar error.
  • Text reported in (2-5) from "author response" is not in the submitted document and cannot be evaluated in the intended context. At the moment, the statement suggested does not seem to be very explicatory. 

A final comment is about the scalability point of view. The authors say: "In terms of scalability, our scheme can hold a large-scale voting by using a huge value of p. For example, our scheme can hold a voting in a town which has 5000 voters and 10 candidates by choosing ?≈10^{5000}". Considering that the largest known prime number is approximately 10^{24,862,047}, probably the system cannot be employed at a national scale. However, a careful and wise reader can deduce this conclusion independently, without other authors' intervention in their text. 

On the contrary, there is no mention of large-scale gas cost or computing time for tallying.
